# Self-Collection for Cervical Screening Programs: From Research to Reality

**DOI:** 10.3390/cancers12041053

**Published:** 2020-04-24

**Authors:** David Hawkes, Marco H. T. Keung, Yanping Huang, Tracey L. McDermott, Joanne Romano, Marion Saville, Julia M. L. Brotherton

**Affiliations:** 1VCS Foundation, Carlton South, VIC 3053, Australia; hkeung@vcs.org.au (M.H.T.K.); whuang@vcs.org.au (Y.H.); tmcdermo@vcs.org.au (T.L.M.); jromano@vcs.org.au (J.R.); msaville@vcs.org.au (M.S.); jbrother@vcs.org.au (J.M.L.B.); 2Department of Pharmacology and Therapeutics, University of Melbourne, Parkville, VIC 3010, Australia; 3Department of Pathology, University of Malaya, Kuala Lumpur 50603, Malaysia; 4Department of Obstetrics and Gynaecology, University of Melbourne, Parkville, VIC 3010, Australia; 5Department of Obstetrics and Gynaecology, University of Malaya, Kuala Lumpur 50603, Malaysia; 6Melbourne School of Population and Global Health, University of Melbourne, Parkville, VIC 3010, Australia

**Keywords:** human papillomavirus, cervical screening, diagnostic testing, self-collection

## Abstract

In 2018, there were an estimated 570,000 new cases of cervical cancer globally, with most of them occurring in women who either had no access to cervical screening, or had not participated in screening in regions where programs are available. Where programs are in place, a major barrier for women across many cultures has been the requirement to undergo a speculum examination. With the emergence of HPV-based primary screening, the option of self-collection (where the woman takes the sample from the vagina herself) may overcome this barrier, given that such samples when tested using a PCR-based HPV assay have similar sensitivity for the detection of cervical pre-cancers as practitioner-collected cervical specimens. Other advantages of HPV-based screening using self-collection, beyond the increase in acceptability to women, include scalability, efficiency, and high negative predictive value, allowing for long intervals between negative tests. Self-collection will be a key strategy for the successful scale up of cervical screening programs globally in response to the WHO call for all countries to work towards the elimination of cervical cancer as a public health problem. This review will examine self-collection for HPV-based cervical screening including the collection devices, assays and possible routine laboratory processes considering how they can be utilized in cervical screening programs.

## 1. Introduction

Cervical cancer is a major public health problem with approximately 570,000 new cases, and 311,000 deaths globally in 2018, the majority of these in low- and middle-income countries [1]. Even within high income countries, most cervical cancers occur in women who are never or under-screened [2,3,4,5]. 

Participation in a screening program involves a woman knowing the reasons for screening, believing that screening is relevant to her, and that being screened provides benefits that outweigh any potential risks or costs. Being able to access a trusted service provider and the ability to undergo a speculum examination are also contributors to women’s decisions about participating in a screening program [6]. Women who live in settings where screening programs are available may be under- or never-screened for a variety of reasons including practical barriers such as work and parenting commitments, financial costs related to attending an appointment for screening, lack of access to appropriate health care services or providers, or other barriers such as a previous negative experience with undergoing a screening test, history of sexual trauma, female genital mutilation, or cultural beliefs [6,7].

Traditional cervical cytology relied on healthcare practitioners visualizing the cervix and sampling cervical cells from the transformation zone of the cervix [8]. Human papillomavirus (HPV)-based cervical screening programs generally use the same sampling approach because HPV positive samples are often then reflexly examined using liquid-based cytology from the same specimen [9]. An advantage of HPV-based testing is that it detects viral nucleic acid, rather than morphological changes in the cell, and as such there is no need to sample from the transformation zone of the cervix because viral nucleic acid is shed from the cervix into the vaginal canal. [10]. Using self-collection, a woman can take the sampling device, insert it into her own vagina to collect a specimen, and then return it to a healthcare provider (either directly or through the mail) without the physical intervention of a healthcare practitioner [11]. This sample can then be processed using a nucleic acid test for the presence of HPV DNA or RNA. In contrast, at present there is not a body of evidence that supports self-collection for morphological analysis (e.g., Pap-stained cells) for the purpose of cervical screening.

Self-sampling has been shown to increase participation in never- or under-screened populations [10]. Several methods for engaging currently non-screening women in cervical screening by utilizing self-collection have been trialed, with varied uptake levels. Pooled analysis of 25 self-collection participation trials showed community campaigns that included community outreach and media support, or door-to-door campaigns, achieved higher participation in the self-collection arms than control arms with practitioner collected cytology or HPV samples, or visual inspection of the cervix with acetic acid by a practitioner (community campaign: 15.6% vs. 6.0%, door-to-door campaign 94.2% vs. 53.0%). Mailing out self-collection kits has also achieved higher participation in the participant arm offered self-collection than the practitioner collection arm (19.2% vs. 11.5%); however, ‘opt in’ trials did not (7.8% vs. 13.4%) [10]. There is some evidence to suggest that face-to-face interactions with a trained healthcare practitioner have the highest likelihood of the largest increases in participation [12].

Self-sampling facilitated cervical screening can be performed in a variety of settings including health care facilities, homes or workplaces facilitated by community health outreach teams [10], with HPV testing of the samples carried out centrally in a laboratory [13,14,15,16,17,18], or on location in the case of Point of Care HPV testing assays [19,20].

This flexibility of self-collection based cervical screening can substantially increase participation in cervical screening in rural, remote or low resource settings [12], and therefore has the potential to increase equity in cervical screening. Modelling by the WHO Cervical Cancer Elimination Modelling Consortium predicts that reducing cervical screening inequity, particularly in low- and middle-income countries, can reduce the burden of cervical cancer at a global level [21]. With the World Health Organization’s call to eliminate cervical cancer as a public health problem by 2120 [22], and the draft elimination strategy target of 70% of the world’s women being screened with a high-performance HPV test by 35 and 45 years of age by 2030 [23], self-collection is likely the only feasible way to scale up and realize this target, given both health workforce constraints and costs, and the limited acceptability to women of speculum-based sample collection in many settings.

In this review, informed by key papers in the literature and our emerging experience in the HPV-based screening program in Australia, we examine a range of different assays and collection devices and consider other pragmatic issues for introducing self-collection into the laboratory as part of an organized HPV-based cervical screening program. We focus not only on emerging research findings but also on how this evidence can be utilized in the paradigm of clinical testing undertaken at pathology laboratories with the sample volumes that could be expected from an organized screening program.

## 2. Materials and Methods

### 2.1. PCR-Based Technologies for HPV Testing of Self-Collected Specimens

There are a large number of HPV testing technologies available but some of the most common are polymerase chain reaction (PCR)-based assays. Major manufacturers such as Abbott, BD, Cepheid, Qiagen, Roche, and Seegene all produce medium- to high-volume, PCR-based HPV assays for use in cervical screening programs. The meta-analysis by Arbyn et al. [10] clearly demonstrated that when self-collected specimens were processed on PCR-based HPV assays, they were as sensitive as clinician-collected specimens for cervical intraepithelial grade 2 or above (CIN2+) across 17 studies, and for CIN3+ across eight studies. There were small but significant reductions in specificity (0.98 (95% CI 0.97–0.99) compared with clinician-collected specimens observed for both CIN2+ and CIN3+. Further analyses [10] also revealed no significant reductions in sensitivity based on the device used for self-collection nor the storage medium. Currently the wealth of evidence supports PCR-based assays for use in self-collection protocols, and this is explicitly stated in the Australian technical requirements for HPV-based cervical screening [24]. It should also be noted that most of the currently utilised medium- to high-volume HPV assays test for the same 14 HPV types; 16, 18, 31, 33, 35, 39, 45, 51, 52, 56, 58, 59, 66, and 68. It is important to note that each different combination of device, buffer and assay/system requires validation, either by the manufacturer or by individual laboratories. A recent study examined dry flocked swabs collected then eluted in ThinPrep media and then tested on six different PCR-based HPV assays, and this was used as the basis for accreditation of this protocol as part of the National Cervical Screening Program in Australia. [25]. As the evidence-base for different combinations of devices and assays grows, there may be refinements to these requirements for component validation, but these are more likely to relate to evidence regarding how stability affects the sensitivity of the assays over time [15]. 

### 2.2. Non-PCR-Based Technologies for HPV Testing of Self-Collected Specimens

#### 2.2.1. Hologic Aptima

There is a wealth of data, including a recent meta-analysis [10], that demonstrate that when self-collected specimens are tested on polymerase chain reaction (PCR)-based diagnostic HPV assays they produce results which are non-inferior to clinician-collected specimens for the detection of underlying high grade cervical lesions (CIN2+). Signal amplification assays such as the Hybrid Capture 2 or Cervista have also been examined but analyses have demonstrated that these types of assays have lower sensitivity for CIN2+ using self-collected specimens than clinician-collected samples [10,26]. The Hologic Aptima assay targets the mRNA of the HPVE6/E7 region of the same 14 types as many of the other clinically validated and automated HPV assays, with a reflex test for partial genotyping of HPV16 and a combined HPV18/45. In a recent meta-analysis, Arbyn et al. [10] examined three studies [27,28,29] which utilised the Aptima HPV assay and determined that, whilst results from self-collected specimens indicated good specificity for cervical intraepithelial neoplasia grade 2 or above (CIN2+), they were significantly less sensitive than clinician-collected samples. Histological results are generally used as the gold standard for assessing cervical screening tests performance with most studies using either CIN2(+) or CIN3(+) as their marker for cervical disease, a precursor to cervical cancer. In the three studies examined by Arbyn and colleagues for the detection of CIN2+ by self-collected specimens tested for HPV on the Aptima, the results indicated that 4/30 [27], 10/69 [28], and 6/16 [29] of histologically-confirmed CIN2+ cases were missed. All 20 of these cases were detected by the paired clinician-collected specimens. 

A recent methodological paper trialled a different protocol for diagnostic testing of self-collected vaginal samples on the Aptima. Borgfeldt and Forslund [30] used an additional heating step (90 °C for 1 h) on specimens stored in the Aptima media. This heating step caused 11/20 previously negative samples to return a positive result on the Aptima assay, and these results were independently confirmed using different assays. When the full cohort of samples (172 specimens) was retested after the novel heating step, the HPV positivity rate increased from 5.3% to 15.9%. This novel protocol raises interesting questions about future optimisation of the Aptima assay for self-collection. It may be worthwhile for the manufacturers to consider further studies, investigating whether the performance of Aptima on self-collected specimens could potentially be improved by the addition of such a heating step. 

Another aspect of the Aptima assay is that it lacks a control for cellularity (sample adequacy), and as a result there is no way of knowing how many of the HPV negative specimens actually contained cellular material. In a cervical screening environment where co-testing (HPV and cytology) are always undertaken, there is no need for a cellularity control as a lack of cellular material is easily identified on cytology. However, with self-collection there is an inherent difficulty in knowing the validity of a negative result unless evidence is available to demonstrate that a sample was actually taken by the woman. Even in clinician-collected cervical screening there is a failure of the cellularity control of between 0.1–0.3% [31,32,33], although inhibition (e.g., by blood or lubricant) can cause the same control failure. There appears to be a growing body of evidence suggesting an increased percentage of invalid specimens (likely due to a lack of endogenous material) in self-collected specimens compared to practitioner-collected samples. [7,25,32,33,34] suggesting some women may agree to self-collect but actually not undertake the test. The presence of common inhibiting reagents, such as lubricants, is likely to be less frequent than when a speculum examination is undertaken. It should be noted that the presence of the endogenous cellularity control does not indicate that specimens have been taken from the cervix or vagina.

The current data appears to suggest that the Aptima assay may not be optimised for self-collection using the current manufacturer’s instructions for use. Further studies using the endpoint of histologically confirmed CIN2/3+ would be useful to confirm the current findings of the Arbyn meta-analysis relating to Aptima [10]. There remains a possibility of further developments to increase the sensitivity of self-collected specimens by altering the extraction protocol. Additionally, the other issue of assay design—the lack of a sample adequacy control—may be resolved by the use of a Hologic-manufactured, research use only cellularity control as is currently in use in Australia.

#### 2.2.2. careHPV

careHPV is an adapted form of the Hybrid Capture 2 assay and was developed as an affordable HPV-DNA detection technology [35,36]. It is specifically designed for low-resource settings, and for potential use as a point of care test, and has been utilised in a variety of geographical locations such as Ghana [36], Bhutan [37], Western Kenya [38] and rural China [35], with diverse demographics including amongst the general population, unscreened women, and women living with HIV [35,36,37]. It should be noted that the suggested cost per test of US$5 may not be achievable [39].

Therefore, is careHPV an appropriate assay for self-collection, when meta-analyses have clearly demonstrated that the Hybrid Capture 2 it was developed from is not [10]? We acknowledge the tension between having an assay that is not cost-prohibitive and able to screen otherwise never-screened women, and the use of an assay that is less sensitive for HPV than a clinician-collected specimen. In general, minor variations in specificity are likely less critical in under- or never-screened populations than reductions in sensitivity. 

There is certainly evidence to suggest that self-collection tested on careHPV facilitates screening by being more acceptable than traditional, speculum-assisted, cervical cytology [37]. A number of studies have been undertaken where there is a lack of a comparator test, either clinician-collected specimen tested on careHPV, or testing of the self-collected specimen on a different HPV assay [37,38], which makes interpretation of the value of the results difficult. A study undertaken in Ghana [36] produced highly concordant results between self- and clinician-collected samples tested on careHPV but did not collect further data to allow assessment of the sensitivity of either test for either cytological or histological markers of disease.

The careHPV assay lacks a control for sample adequacy, or for inhibition (such as by blood). The meta-analysis by Arbyn [10] determined that careHPV was significantly less sensitive for HPV associated with CIN2+ lesions when a self-collected sample was used rather than a clinician-collected sample. The current state of data indicates that careHPV, whilst being cheaper and therefore more accessible, has significantly lower sensitivity for HPV from self-collected specimens and is therefore unlikely to be appropriate for cervical screening of these samples because of the risk of cases of disease being missed. This is a significant issue in settings where women may only receive a once- or twice-in-a-lifetime screening.

### 2.3. Self-Collection Devices for HPV Testing

#### 2.3.1. Evalyn Brush

Of the current devices in use for self-collection, the Evalyn brush (Rovers Medical Devices, Lekstraat, The Netherlands) is possibly the most established and evidence-based device. It is currently used in a number of jurisdictions, including as the device used in the self-collection pathway of The Netherlands national cervical screening program. The Evalyn brush is marketed as a customized device designed to be used for self-collection for HPV-based cervical screening. 

There are a wide range of studies examining the use of Evalyn brush in detecting 14 oncogenic HPV genotypes on multiple clinically validated PCR-based HPV assays, including the Roche cobas 4800 [13,14,15], Abbott RealTime [16,17], BD Onclarity [18], Cepheid Xpert [15], Seegene Anyplex II [15], GP5+/6+ PCR enzyme immunoassay [40,41,42], and SPF10 PCR-DEIA-LiPA25 [43,44]. These studies were undertaken across a range of developed countries including China [17], Germany [16], Netherlands [13,40,41], Denmark [14], Norway [15], and Spain [42].

In the Arbyn meta-analysis [10] there were four studies using the Evalyn brush, one tested on the HC2, and three on validated, PCR-based tests. When an Evalyn brush self-collected specimen was tested on the HC2 [45], it had a relative sensitivity compared with clinician-collected specimens of 0.66. When the Evalyn brush self-collected samples were tested on PCR-based assays, they had a relative sensitivity of 0.99. These data suggest that, in most cases, it is likely to be the type of assay, rather than the collection device, that will predict the clinical quality of the result.

An interesting study published by a Norwegian group [17] examined the sensitivity and specificity of the Evalyn brush and a Copan FLOQswab (Copan, Brescia, Italy) compared to clinician-collected samples tested on three PCR-based assays: Cepheid Xpert, Seegene Anyplex II, and the Roche cobas 4800 HPV test. Relative sensitivity, compared to clinician-collected specimens, for CIN3+ demonstrated that the Copan swab was significantly less sensitive than the clinician-collected across all three assays, but that the Evalyn brush was not. The authors presented a further analysis which segregated the results based on the time between collection and specimen preparation (stabilization). If the time between specimen collection and preparation was 28 days or less, there was no longer a significant drop in sensitivity for the Copan swab for CIN3+. There was also variation in which the self-collection device was more sensitive depending on the assay used. Another study [15], comparing the sensitivity for CIN2+ of the Evalyn brush with the Qvintip device (Aprovix, Stockholm, Sweden), found no significant differences between either of the self-collection devices or the clinician-collected sample. For longer intervals between collection and preparation, the Evalyn brush showed similar levels of sensitivity which suggests that this device may be superior when extended delays between collection and processing cannot be avoided [15]. The major concern with using the Evalyn brush, particularly in low- or middle-income countries, is the cost. Prices obviously vary depending on local conditions but the cost of the Evalyn brush appears to be between 3–5 times the price of other self-collection devices, such as the Copan FLOQswab. 

#### 2.3.2. Viba-Brush

As highlighted above, the one major drawback of the Evalyn brush is its cost. Rovers Medical Devices have sought to address this issue by introducing a more affordable version of the Evalyn collection device called the Viba-brush. The Viba-brush utilises the same collection head but mounts it more simply on a straight handle and is reported to have a price equivalent to other low-cost devices. 

There are two paired sample studies [43,44] which examined the relative sensitivity of HPV tests performed on self-collected samples using the Viba-brush compared with clinician-collected specimens using CIN2+ as the clinical endpoint.

In the Dijkstra study [44] undertaken in the Netherlands, 43/135 recruited women had biopsy-confirmed CIN2+. The self-collection had a sensitivity of 93.0% (40/43) and clinician-collection had a sensitivity of 90.7% (39/43) using a PCR-based assay. Two CIN 2 lesions were hrHPV negative in both sample types. 

In the study undertaken by Geraets et al. [43] biopsies were undertaken from 49 women (out of a cohort of 182) after evidence of an abnormality was discovered during colposcopy. Two HPV assays were used to test all clinician- and self-collected specimens. The clinician-collected specimens were more sensitive for CIN2+ than the self-collected samples on both assays (Table 1.)

#### 2.3.3. Qvintip

There are only a few studies in the past ten years which examine the use of the Qvintip device for HPV testing. Four of these studies [46,47,48,49] are from the same research group where a senior member was a minority shareholder in the company that produces Qvintip. These studies do provide good information on the acceptability of self-collection using the Qvintip device but lack controls with none comparing paired samples, either two self-collected or one self-collected and one clinician-collected specimens. Follow-up (cytology, histology) was only undertaken on HPV positive participants so sensitivity is not calculable.

There is a direct comparison study by Jentschke and colleagues [16] which provide data that suggest that the Qvintip may be easier to use and less likely to cause discomfort than the Evalyn brush. These data also suggest, although this is not a statistically significant result, that the Qvintip is less sensitive for CIN2+ (83.7%) compared with either the Evalyn brush (89.8%) or a clinician-collected specimen (89.8%). The differences in both ease of use and sensitivity are small and it is possible that these differences may stem from the different instructions for use; the Evalyn brush was inserted into the vagina and rotated five times before being removed, whereas the Qvintip was simply inserted then removed without any rotation. Interestingly, the Qvintip instructions for use [50] now indicate that it should be rotated after insertion into the vagina.

#### 2.3.4. Copan FLOQSwab

Another option for self-collection being increasingly used in cervical screening programs is the flocked swab, generally the FLOQswab made by Copan (Brescia, Italy). There are a few studies that examined self-sampling using flocked swabs, but it is difficult to assess these studies together as the specific type of FLOQswab used is not always described. A small study [51] examining the Copan self-collection swab compared with a cotton swab determined that the flocked swab was more sensitive for HPV16, HPV18 and for other high-risk HPV types. Although numbers of histologically confirmed CIN2+ were low, there was a tendency for better detection by the flocked sample (9 CIN2+) compared with the cotton swab (5 CIN2+). Another small study [52] used the Copan ESwab™ for self-collection and compared it to a self-collection sample stabilized on an FTA^®^-cartridge, and a clinician-collected specimen. The flocked swab had equal or greater sensitivity than both the FTA^®^-cartridge and the clinician-collected specimen. The flocked swab also matched the clinician–collected sample for detection of cytological high-grade squamous intraepithelial (HSIL) lesions and cervical cancers and was more sensitive than the FTA^®^-cartridge method. Both of these studies used the clinically-validated Seegene Anyplex HPV assay. 

The Copan flocked swab (product number #552) has also be used in the national cervical screening programs in both Malaysia and Australia after successful pilot implementation studies [7,20]. 

### 2.4. Other Considerations for Clinical Testing of Self-Collection Specimens for HPV

#### 2.4.1. Validated Assays

It is emerging best practice for the use of HPV assays in screening to require the assay to have evidence of being clinically validated, with the two major mechanisms for assessing clinical sensitivity and specificity being the Meijer Criteria [53] or VALGENT [54]. The Meijer Criteria [53] examines the assay’s clinical sensitivity and specificity (for histologically confirmed CIN2+) compared with a reference assay, with an additional group of HPV positive enriched specimens for examining intra- and inter-laboratory reproducibility. VALGENT (VALidation of HPV GENotyping Tests) [54] uses a slightly different protocol for assessing specificity and sensitivity compared to a reference assay. It requires 1300 consecutive screening specimens supplemented with 300 abnormal cytology specimens and HPV results, along with follow-up histology where appropriate to determine relative sensitivity and specificity for CIN2+.

Both The Netherlands and Australia have self-collection pathways as part of a HPV-based national cervical screening program, with The Netherlands using the Evalyn brush and laboratories in Australia who are accredited currently all utilizing the Copan FLOQSwab. Currently only Roche cobas HPV assays (cobas or cobas 4800) are being used in these national programs for self-collected specimens. In Australia the availability of self-collection was delayed by lack of any suitable clinically validated HPV assays having manufacturer validated claims for the use of self-collected specimens (on label use), which required laboratories to independently validate their self-collection protocols against paired clinician-collected specimens ([25], other studies not published).

Whilst a number of assays present data from the scientific literature indicating that their assays perform well with self-collected specimens, at the time of writing, there does not appear to be any clinically validated assays with widely accepted (e.g., CE mark or FDA-approved) instructions for use covering self-collected specimens.

#### 2.4.2. Sample Preparation

One of the less commonly addressed issues with self-collection is the actual processing of the sample within an accredited pathology laboratory, as opposed to a research laboratory. Many of the research studies investigating self-collection involve the self-collected specimen being transported to the laboratory as a dry sample where it is then resuspended in a liquid media [7,13,16]. The mechanism by which these dry samples are resuspended is highly varied. The most common buffers used are liquid-based cytology media, such as Hologic PreservCyt or BD SurePath, as these solutions are validated for use with HPV assays. Different volumes are used in different studies and often the mechanism by which the dry sample is eluted into these solutions is poorly described. At present, there is no consensus protocol for sample preparation, but this is likely to change if any of the major assay manufacturers add a self-collection claim to their assays. Use of such a manufacturer’s protocol would simplify pathology laboratory accreditation for self-collection.

Another important consideration in the resuspension of dry samples is the availability of media. Many countries do not have stable supplies of liquid-based cytology media and even transporting these media can be difficult, as they are alcohol-based and flammable. This means that they must be shipped as a dangerous good (in volumes > 300 mL), increasing costs which may be prohibitive for low- and middle-income countries. Some manufacturers are looking to validate non-alcohol-based media to reduce these issues, although this may in turn result in reduced stability which could restrict use to point of care testing. 

#### 2.4.3. Pre-Analytic Considerations

In addition to a lack of validated HPV assays for self-collection and a consensus elution protocol, another issue for future scale up of its use is that self-collection is currently dependent on a number of manual steps. If self-collection is going to become widely available, and laboratories are going to process hundreds of samples a day, there will be a need for pre-analytic automation—both for workflow and for the reduction in inter-operator error. Current automated pre-analytic instruments, such as the Roche p480, do not have the ability to process self-collected devices. 

Another possible mechanism would be to elute dry swabs directly into a solution already used for the processing of specimens, such as the BD diluent or the Roche cobas PCR media. There are currently two studies [55,56] which utilised the Roche cobas PCR media as the buffer into which a dry swab was eluted. This methodology (sample in Roche cobas PCR media) could be suitable using the Roche p480 pre-analytic instrument. Both studies transferred the swab directly into the cobas PCR media at the point of collection, and as such there are currently no data on the stability of a swab transported dry to a laboratory before being resuspended into this media. BD diluent has been used as buffer, but only in an ad hoc method rather than using the commercially available diluent tubes [57].

## 3. Discussion

The updated meta-analysis on the validity of using self-collection methods for HPV testing in 2018 by Arbyn et al. [10] was of critical importance as it demonstrated that the body of evidence had grown over the four years since the initial meta-analysis to a point where it could be clearly demonstrated that PCR-based assays offered similar levels of sensitivity for HPV in self-collected specimens compared with clinician-collected specimens. All 11 assays included in the analysis had self-collection sensitivities for HPV that were not significantly lower than the clinician-collected samples. 

This review also aimed to summarize the current evidence on the utility of other technologies in national screening programs. careHPV presents as a viable test because of the very low cost, although the exact cost per test appears to be in question [39]. The data presented in the review supports the analysis by Arbyn et al. [10], that the lack of sensitivity of careHPV is likely to result in false negative results for specimens from women with CIN2+. 

The Hologic Aptima HPV test uses RNA as its template, which may potentially give better specificity which could be beneficial for managing limited follow-up resources such as colposcopy more effectively. There are only a relatively small number of studies using Aptima and, because of the different mechanism of detection of HPV results compared to other assays, results cannot be pooled with PCR-based tests. All studies comparing self-collection on Aptima indicate that it is a less sensitive test for CIN2+ which suggests that, if there is higher specificity, it is to the detriment of detecting disease.

The evidence supporting non-PCR-based HPV tests for use with self-collected samples is not as favourable or as comprehensive as that supporting PCR-based testing. Further evidence on the application and performance of non-PCR-based technologies for self-sampling are likely to emerge and any new information will be critical for any re-assessment of use of these technologies for self-collection.

In contrast to the importance of correct HPV assay selection for the testing of self-collected specimens, a wide range of vastly different sampling devices appear to produce good results. The Arbyn meta-analysis [10] covered 10 different self-sampling devices and none produced significantly lower sensitivity than the clinician-collected specimen. This review examined a number of these devices in detail. The information seems to suggest that factors other than clinical sensitivity may be the deciding factor for the use of a device within a specific cervical screening program. The cost of self-collection devices can be prohibitive, with the FTA^®^-cartridge costing more than US$5 whereas a Copan flocked swab is under US$1 [52]. Environmental stability is also a variable requirement depending on the situation—temperature and humidity vary greatly from region to region. Another aspect of a self-collection device is whether it can be modified to high volume throughput. Few of the current device/assay combinations are designed for medium to high throughput (>200, and >1000 clinical specimens a day respectively), the exception being the cobas swab shipped in cobas PCR media. As we move towards self-collection as a primary cervical screening pathway, throughput will have to be considered, and incorporated into the design of the next general of pre-analytic automation.

Another emerging methodology for self-collection is the use of urine for HPV-based cervical screening. This particular method was not covered in this review as there is a paucity of data on the predictive value of a urine HPV positive result as it relates to clinical outcomes such as histologically—confirmed high-grade cervical disease. There is currently a diagnostic test accuracy study being undertaken called Validation of Human Papillomavirus Assays and Collection Devices for Self-samples and Urine Samples (VALHUDES) [58], which seeks to examine the clinical sensitivity and specificity of urine, and vaginal self-collected samples, against both a clinician-collected samples and against the histological diagnosis of followed up cases. It is likely that VALHUDES will address a number of other issues surrounding the development of a standardized urine collection and testing protocol [59], although this trial utilises a specific urine collection device, the Colli-Pee (Novosanis NV, Wijnegem, Belgium), which may be cost prohibitive in low- and middle-income countries.

There is a need for further research into the acceptance and feasibility of using self-collection in diverse under- or never-screened populations, but this falls outside the scope of the current review which is more focused on the technical aspects of self-collection.

## 4. Conclusions

This review sought to examine how the current evidence base could be used to support self-collection as a method of increasing cervical screening. There were three key findings relating to what type of test should be used, what type of collection devices should be used and what factors need to be considered moving forwards:Self-collection is an attractive mechanism to increase participation in cervical screening worldwide. The current evidence strongly supports the need for PCR-based HPV assays with internal controls, specifically for both sample adequacy and detection of inhibition.Whilst there does not appear to be major differences in the sensitivity of different collection devices, their cost, acceptability, and scalability as part of population level screening programs need to be considered.This area of clinical testing for HPV as part of organized cervical screening programs will likely be a fast evolving one, with the continued development of new HPV assays, pre-analytic devices, and hopefully, manufacturer-validated claims for the use of self-collection.

## Figures and Tables

**Table 1 cancers-12-01053-t001:** Sensitivity for CIN2+ of clinician-collected specimens and self-collection using Viba-brush specimens.

Specimen Type	GP5+/6+-EIA PCR Assay [43]	SPF10 PCR-DEIALiPA25 PCR Assay [43]	GP5+/6+-EIA PCR Assay [44]
Clinician-collected	48/49 (98.0%)	49/49 (100%)	39/43 (90.7%)
Self-collected	43/49 (87.8%)	47/49 (95.9%)	40/43 (93.0%)

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
