# Peer review of "Self-Collection for Cervical Screening Programs: From Research to Reality"

_cancers, 2020, doi:10.3390/cancers12041053_

Round 1

Reviewer 1 Report

Hawkes et al present a piece on self sampling for cervical screening purposes. The manuscript is topical and timely.  It also benefits from being created by a team expert and practiced  in the implementation and delivery of HPV based screening and the evaluation of emerging technologies to support it. I enjoyed reading it

This said, there are some issues with the manuscript which I feel if addressed could improve it further. I accept that it is not a systematic review on the performance of HPV based self sampling however some parts provide/refer to published data in great detail while others tend towards own-opinion based on a single article or a “general acknowledgement” which creates a slightly uneven tone. A general suggestion is that the length could be reduced by 20-25% without losing the essential points

Specific comments

Line 65 – what is meant by the “control group” here is not immediately clear

Line 105 – There is s slightly brief statement on appropriate validation of self sampling devices. This is an important area and would benefit from expansion.  While I totally accept that the meta-analysis show little difference in performance according to device, external accreditation agencies require validation of a particular system/buffer/device internally at the very least. How this may be approached is important.

Table 1 – Why is kappa used? Arguable this inflates the level of agreement particularly in sample-sets where the majority of samples are test-negative. The levels of agreement that kappa provide (“good” etc) are also arbitrary and not necessarily transferrable according to what is acceptable for screening.  Could results be presented by percent positive agreement and percent negative agreement…Also where are the 95% CI’s around the kappas ? If the authors are using this table to make statement about differences then more detailed information on the statistical weight/size of these differences  need to be provided.

Lines 148-153  These lines are quite detailed and  - as per earlier comment – the m/s sometimes veers from great detail to broad brush statements. The section on APTIMA performance occasionally contradicts itself  - Toliman appeared to show little difference in detection of HSIL across assays (for example) but the conclusion is that APTIMA is suboptimal as a test. Suggest a reconsideration (and potentially a shortening) of the APTIMA section to convey the main points.

Line 172 – I do not agree with the authors that there is a “general acknowledgement” that self samples have significantly higher rates of endogenous control failure.  The one study which is cited in the m/s as demonstrating a 10% invalidity rate is not mirrored in other studies/exercises.  Suggest that the authors relax this statement or better qualify it.  Also the authors, arguably, do not describe the limitations of endogenous controls as a control for sample adequacy given that the do not indicate the presence of relevant cervical epithelial cells

Lines 202-214: Repetitious and could be summarised

Line 290 – Spell out FTA acronym

Line 315 – Care has a claim for self collection as I understand it. I agree with the authors that the clinical performance may be suboptimal though!

A nit pick – but VALHUDES is not a clinical trial in the true sense given that there is no randomisation and results do not affect pathways. Clinical study would be a more accurate term

The use of the phrase “not promising” for transcription media amplification is maybe a little broad brush again. Particularly as the authors rightly allude to the fact that Forsulnd et al have shown the pre analytical processing may confer a positive impact for self samples tested using this technology. An alternative phrase would/could be that further evidence on the application and performance of  non PCR based technologies for self sampling are likely to emerge in the future and that this will be important.

Author Response

Hawkes et al present a piece on self sampling for cervical screening purposes. The manuscript is topical and timely.  It also benefits from being created by a team expert and practiced  in the implementation and delivery of HPV based screening and the evaluation of emerging technologies to support it. I enjoyed reading it

This said, there are some issues with the manuscript which I feel if addressed could improve it further. I accept that it is not a systematic review on the performance of HPV based self sampling however some parts provide/refer to published data in great detail while others tend towards own-opinion based on a single article or a “general acknowledgement” which creates a slightly uneven tone. A general suggestion is that the length could be reduced by 20-25% without losing the essential points

Thank you for this feedback. We have reduced the manuscript length in the sections noted in the following responses to the reviewers (e.g. removal of original Table 1).

Specific comments

Line 65 – what is meant by the “control group” here is not immediately clear

This has now been clarified to explain that it means practitioner collected using one of a variety of screening methods dependent upon the study

Line 105 – There is s slightly brief statement on appropriate validation of self sampling devices. This is an important area and would benefit from expansion.  While I totally accept that the meta-analysis show little difference in performance according to device, external accreditation agencies require validation of a particular system/buffer/device internally at the very least. How this may be approached is important.

We thank the reviewer for highlighting this point. The following statement has been added

‘It is important to note that each different combination of device, buffer and assay/system requires validation, either by the manufacturer or by individual laboratories. A recent study examined dry flocked swabs collected then eluted in ThinPrep media and then tested on six different PCR-based HPV assays and this was used as the basis for accreditation of this protocol as part of the National Cervical Screening Program in Australia. [25].’

Table 1 – Why is kappa used? Arguable this inflates the level of agreement particularly in sample-sets where the majority of samples are test-negative. The levels of agreement that kappa provide (“good” etc) are also arbitrary and not necessarily transferrable according to what is acceptable for screening.  Could results be presented by percent positive agreement and percent negative agreement…Also where are the 95% CI’s around the kappas ? If the authors are using this table to make statement about differences then more detailed information on the statistical weight/size of these differences need to be provided.

Rather than develop and explain this table further, which contained data extracted from the published studies cited, this Table has been removed as it appears to have been a source of confusion and did not appear to add significant value to the manuscript.

Lines 148-153  These lines are quite detailed and  - as per earlier comment – the m/s sometimes veers from great detail to broad brush statements. The section on APTIMA performance occasionally contradicts itself  - Toliman appeared to show little difference in detection of HSIL across assays (for example) but the conclusion is that APTIMA is suboptimal as a test. Suggest a reconsideration (and potentially a shortening) of the APTIMA section to convey the main points.

We thank the reviewer for their comment. We have removed the detail and edited some of the text from the Aptima section, removing the recent studies with non-histological outcomes.

Line 172 – I do not agree with the authors that there is a “general acknowledgement” that self samples have significantly higher rates of endogenous control failure.  The one study which is cited in the m/s as demonstrating a 10% invalidity rate is not mirrored in other studies/exercises.  Suggest that the authors relax this statement or better qualify it.  Also the authors, arguably, do not describe the limitations of endogenous controls as a control for sample adequacy given that the do not indicate the presence of relevant cervical epithelial cells

We thank the author for their input. We have modified the text and added a number of studies based in Australia that highlight the invalid rate for HPV-based cervical screening as well as the higher invalid rate for a pilot of the self-collection based cervical screening method used in Australia. These published data are also consistent with observed invalid rates in a laboratory testing self-collected specimens in the Australian National Cervical Screening Program (unpublished data).

‘There appears to be a growing body of evidence suggesting an increased percentage of invalid specimens (likely due to a lack of endogenous material) in self-collected specimens compared to practitioner-collected samples. [7,25,35-37] suggesting some women may agree to self-collect but actually not undertake the test. The presence of common inhibiting reagents, such as lubricants, is likely to be less frequent than when a speculum examination is undertaken. It should be noted that the presence of the endogenous cellularity control does not indicate that specimens have been taken from the cervix or vagina.’

Lines 202-214: Repetitious and could be summarised

The paragraph has been edited for brevity.

‘The careHPV assay lacks a control for sample adequacy, or for inhibition (such as by blood). The meta-analysis by Arbyn [10] determined that careHPV was significantly less sensitive for HPV associated with CIN2+ lesions when a self-collected sample was used rather than a clinician-collected sample The current state of data indicates that careHPV, whilst being cheaper and therefore more accessible, has significantly lower sensitivity for HPV from self-collected specimens and is therefore unlikely to be appropriate for cervical screening of these samples because of the risk of cases of disease being missed. This is a significant issue in settings where women may only receive once or twice in a lifetime screening’

Line 290 – Spell out FTA acronym

FTAâ is a registered trademark in itself. It is derived from Flinders Technology Associates. The trademark symbol has been added throughout the document.

Line 315 – Care has a claim for self collection as I understand it. I agree with the authors that the clinical performance may be suboptimal though!

Whilst a number of studies have examined careHPV, there does not appear to be any mention of self-collection in the careHPV instructions for use or in the careHPV WHO pre-qualification documents. We have left the manuscript unchanged.

A nit pick – but VALHUDES is not a clinical trial in the true sense given that there is no randomisation and results do not affect pathways. Clinical study would be a more accurate term

VALHUDES is listed on clinicaltrials.org as a ‘diagnostic test accuracy study’ and this terminology has been incorporated into the manuscript.

The use of the phrase “not promising” for transcription media amplification is maybe a little broad brush again. Particularly as the authors rightly allude to the fact that Forslund et al have shown the pre analytical processing may confer a positive impact for self samples tested using this technology. An alternative phrase would/could be that further evidence on the application and performance of  non PCR based technologies for self sampling are likely to emerge in the future and that this will be important.

The paragraph mentioned by the reviewer has been rewritten;

‘The evidence supporting non-PCR-based HPV tests for use with self-collected samples is not as favourable or as comprehensive as that supporting PCR-based testing. Further evidence on the application and performance of non-PCR-based technologies for self sampling are likely to emerge and any new information will be critical for any re-assessment of use of these technologies for self-collection.’

Reviewer 2 Report

Dear authors,

Overall this was a clearly articulated and comprehensive summary of the current evidence on methods available for self-collection of HPV cervical screening programs. Given the vast inequities between and within countries and the World Health Organisation's elimination target this is a very timely and important piece of work.

Below are my specific comments, most of which are related to minor typographical, grammatical or formatting issues.

  • Commas – There are some unnecessary commas throughout, for e.g. Line 39, 40 (there are only two things in the list, so you do not need a comma before the “and”) and Line 60 (there are only two things in the list, so you do not need a comma before the “or”). You also sometimes use the Oxford comma (e.g. line 75) and you sometimes don’t (e.g. line 71) – perhaps be consistent. At Line 84 you either need to add an “and”, so that it reads “different assays and collection devices”, as there are only two items in the list. Alternatively, you can remove the word “consider”, so that there are then three points included in this phrase.
  • This review appears to be very comprehensive, both in breadth of scope and the depth of literature you have reviewed. I appreciate, from the lack of any information to say otherwise, that this was not a systematic review. I still however, think that some description of the method used to identify and summarise relevant literature is warranted. It may only be a paragraph, but I think readers need something so assess if whether the evidence they are being presented with here is telling the whole story/state of the evidence, or whether there may be some bias introduced here, i.e. is there any perspective that is likely not included here.
  • Line 132/138 and throughout – You may want to consider spelling out numbers if they start the sentence (I personally think it is an outdated grammatical rule).
  • Table 1 – I think this needs to be better formatted. Firstly, a border and some gridlines would make it look better, and potentially easier to read. As Tables should be able to stand alone, you need to state in the table what your strata are (e.g. by type of assay) and what is being presented in each of the columns (e.g. study and kappa statistic). In text (and potentially as a footnote in the table) you should give the reader some indication of kappa cut-offs to aid with the interpretation.
  • Line 176 –What does IFU stand for; please spell it out.
  • Line 303 – I think it is worth explaining what the Meijer Criteria and VALGENT are, either in text or add an information box that you can refer to. This will help with the understanding of this section for readers who may not be directly in, but work adjacent, to this specific clinical and research area. Also, please spell out the VALGENT acronym the first time it is mentioned.
  • Line 261-263 - I was interested in the unreported data that you refer to here. In every other section you talk through the relevant literature, but you don’t here. Is there a reason for this? Can you summarise it in a table?
  • Line 310 – can you please reference these Australian clinical trials here (perhaps consider summarising them in a table?)
  • Results – please consider identifying the main research question and summarising the findings in a table. While you have mostly done an excellent job at doing this in text, tables can be very useful – especially for the time-poor reader!
  • Line 352 – As this is a new paragraph – and a new section – give some indication of what the updated or original meta-analysis was on. For example, “The updated meta-analysis on the validity of using self-collection methods for HPV testing in 2018 by Arbyn et al. was…”
  • As you are using a numbering system for references (not listing in alphabetic order by author name), you should add in the citation for Arbyn et al. 2018 the first time you refer to this paper in this section.
  • Line 358-359 – This first sentence is worded awkwardly ( “…this review had were about how..” – I had to read this part a few times). Perhaps, “This review also aimed to summarise the current evidence on the utility of other technologies in national screening programs”.
  • Line 359-360 – I also found the line “…although this claim appears to be in question” unclear. It appears that careHPV is not as cost-effective as initially claimed – the cost-effectiveness, not the claim, is in question.
  • Line 361-362 – I believe you are missing a word in this sentence (in brackets here), “..the lack of sensitivity of careHPV is likely (to) result in false negative results for specimens from women with CIN2+”
  • Line 365 – “There remain only a relatively small number of studies using Aptima…”. What do you mean by ‘remain’ in this sentence? Were some studies removed/deleted/not rigorous enough?
  • Line 381 – the point about temperature and humidity is one I had wondered about reading your review & I had hoped for a greater discussion of it here. Is there any evidence at all of the impact of ambient climate on the accuracy of the tests, etc? Are there any expert opinion on the impact of ambient temp/humidity on self-collection that you could refer to here? You say it is “certainly acknowledged” – do you mean by you, or by experts in the field? If the latter, can you cite any of their work or hypotheses?
  • Line 382/384 - Excuse my ignorance by what is meant by “high volume throughput”. This was not bought up in the results/synthesis of your review and it seems like a random addition. Perhaps if you give the reader a little more information, the relevance will become apparent to those who work in adjacent fields.
  • Line 391 – if VALHUDES is an acronym, can you spell it out first.
  • Discussion – I think there is room in the discussion to briefly talk to the importance of self-collection in reducing within-country disparities in cervical screening and cervical cancer rates, and that the design and implementation of self-sampling programs/approaches need to be under-pinned by evidence on how well methods, tests, devices, approaches, etc work in the never and under-screened populations. Using evidence from the general screen-eligible population may not lead to an equitable program, and may further widen current disparities.
  • Throughout (eg. Line 352, Line 361) - there needs to be a period after al in et al. (Arbyn et al. was…)
  • Throughout – you sometimes hyphenate the words self-collect, self-collected, and self-collection, whereas sometimes you leave out the hyphen. APA style advises that “self-“ words should always be hyphenated.
  • Throughout – compound adjectives preceding the term that they modify, should be hyphenated. You have not always done this, e.g. evidence-based devise (line 219). Compound adjectives that do not precede the modifying term do not need to be hyphenated. My preferences is that they are for consistency. But as long as you are consistent throughout, it does not matter. Perhaps check, line 270 follow-up, 373 flip-side, and throughout.
  • The formatting for the in-text citations is not correct on lines 72, 272.
  • Check formatting of references. E.g. reference 10 – HPV should be capitalised.

Author Response

Dear authors,

Overall this was a clearly articulated and comprehensive summary of the current evidence on methods available for self-collection of HPV cervical screening programs. Given the vast inequities between and within countries and the World Health Organisation's elimination target this is a very timely and important piece of work.

Below are my specific comments, most of which are related to minor typographical, grammatical or formatting issues.

  • Commas – There are some unnecessary commas throughout, for e.g. Line 39, 40 (there are only two things in the list, so you do not need a comma before the “and”) and Line 60 (there are only two things in the list, so you do not need a comma before the “or”). You also sometimes use the Oxford comma (e.g. line 75) and you sometimes don’t (e.g. line 71) – perhaps be consistent. At Line 84 you either need to add an “and”, so that it reads “different assays and collection devices”, as there are only two items in the list. Alternatively, you can remove the word “consider”, so that there are then three points included in this phrase.

Thank you. We have made the suggested edits and reviewed the document for consistency.

  • This review appears to be very comprehensive, both in breadth of scope and the depth of literature you have reviewed. I appreciate, from the lack of any information to say otherwise, that this was not a systematic review. I still however, think that some description of the method used to identify and summarise relevant literature is warranted. It may only be a paragraph, but I think readers need something so assess if whether the evidence they are being presented with here is telling the whole story/state of the evidence, or whether there may be some bias introduced here, i.e. is there any perspective that is likely not included here.

We have added the following to the last paragraph of the introductory section:

‘In this review, informed by key papers in the literature and our emerging experience in the HPV based screening program in Australia, we examine a range….’

  • Line 132/138 and throughout – You may want to consider spelling out numbers if they start the sentence (I personally think it is an outdated grammatical rule).

Following revisions, no sentences now start with numbers.

  • Table 1 – I think this needs to be better formatted. Firstly, a border and some gridlines would make it look better, and potentially easier to read. As Tables should be able to stand alone, you need to state in the table what your strata are (e.g. by type of assay) and what is being presented in each of the columns (e.g. study and kappa statistic). In text (and potentially as a footnote in the table) you should give the reader some indication of kappa cut-offs to aid with the interpretation.

This Table has been removed as above.

  • Line 176 –What does IFU stand for; please spell it out.

The text is ‘using the current manufacturer’s instructions for use (IFU)’. However we have now deleted IFU to avoid this confusion, as the abbreviation is not subsequently used.

  • Line 303 – I think it is worth explaining what the Meijer Criteria and VALGENT are, either in text or add an information box that you can refer to. This will help with the understanding of this section for readers who may not be directly in, but work adjacent, to this specific clinical and research area. Also, please spell out the VALGENT acronym the first time it is mentioned.

The following text has been added;

‘The Meijer Criteria [56] examines the assay’s clinical sensitivity and specificity (for histologically confirmed CIN2+) compared with a reference assay, with an additional group of HPV positive enriched specimens for examining intra- and inter-laboratory reproducibility. VALGENT (VALidation of HPV GENotyping Tests) [57] uses a slightly different protocol, for assessing specificity and sensitivity compared to a reference assay. It requires 1300 consecutive screening specimens supplemented with 300 abnormal cytology specimens and HPV results, along with follow-up histology where appropriate to determine relative sensitivity and specificity for CIN2+.’

  • Line 261-263 - I was interested in the unreported data that you refer to here. In every other section you talk through the relevant literature, but you don’t here. Is there a reason for this? Can you summarise it in a table?

The Viba-brush section has been modified in line with the reviewer’s suggestions

  • Line 310 – can you please reference these Australian clinical trials here (perhaps consider summarising them in a table?)

There are currently two laboratories in Australia validated to undertake self-collection for HPV-based cervical screening. One of these studies is now accepted (Reference Saville, Hawkes et al, Journal of Clinical Virology) but the other is not. This information has been added to the text.

  • Results – please consider identifying the main research question and summarising the findings in a table. While you have mostly done an excellent job at doing this in text, tables can be very useful – especially for the time-poor reader!

 We thank the reviewer for this suggestion and have modified the Conclusion to be easier to read.

  • Line 352 – As this is a new paragraph – and a new section – give some indication of what the updated or original meta-analysis was on. For example, “The updated meta-analysis on the validity of using self-collection methods for HPV testing in 2018 by Arbyn et al. was…”

We thank the reviewer and have made the recommended change;

‘The updated meta-analysis on the validity of using self-collection methods for HPV testing in 2018 by Arbyn et al. [10] was of critical importance as it demonstrated that the body of evidence had grown over the four years since the initial meta-analysis to a point where it could be clearly demonstrated that PCR-based assays offered similar levels of sensitivity for HPV in self-collected specimens compared with clinician-collected specimens.’ 

  • As you are using a numbering system for references (not listing in alphabetic order by author name), you should add in the citation for Arbyn et al. 2018 the first time you refer to this paper in this section.

The citation for the Arbyn et al. 2018 paper has been added.

  • Line 358-359 – This first sentence is worded awkwardly ( “…this review had were about how..” – I had to read this part a few times). Perhaps, “This review also aimed to summarise the current evidence on the utility of other technologies in national screening programs”.

The reviewer’s recommendation alteration to the text has been made.

  • Line 359-360 – I also found the line “…although this claim appears to be in question” unclear. It appears that careHPV is not as cost-effective as initially claimed – the cost-effectiveness, not the claim, is in question.

The above phrasing has been altered to “although the exact cost per test appears to be in question” in order to clarify the topic.

  • Line 361-362 – I believe you are missing a word in this sentence (in brackets here), “..the lack of sensitivity of careHPV is likely (to) result in false negative results for specimens from women with CIN2+”

 We thank the reviewer and have made the suggested correction.

  • Line 365 – “There remain only a relatively small number of studies using Aptima…”. What do you mean by ‘remain’ in this sentence? Were some studies removed/deleted/not rigorous enough?

We have replaced the word ‘remain’ with ‘are’ and edited this sentence.

  • Line 381 – the point about temperature and humidity is one I had wondered about reading your review & I had hoped for a greater discussion of it here. Is there any evidence at all of the impact of ambient climate on the accuracy of the tests, etc? Are there any expert opinion on the impact of ambient temp/humidity on self-collection that you could refer to here? You say it is “certainly acknowledged” – do you mean by you, or by experts in the field? If the latter, can you cite any of their work or hypotheses?

We agree with the reviewer that this is a salient point for self-collection and have included a sentence that highlights both the lack of data as well as some unpublished data from our laboratory

‘There is not a lot of data currently published examining temperature and humidity but one Australian Laboratory (VCS Pathology) has demonstrated sensitivity and stability of low positive (~3XLOD) HPV specimens on Copan FLOQSwabs for 28 days at 50Ëš Celsius and 100% relative humidity (unpublished data).’

  • Line 382/384 - Excuse my ignorance by what is meant by “high volume throughput”. This was not bought up in the results/synthesis of your review and it seems like a random addition. Perhaps if you give the reader a little more information, the relevance will become apparent to those who work in adjacent fields.

The sentence has been altered to define medium and high-throughput as >200 and >1000 clinical specimens a day

  • Line 391 – if VALHUDES is an acronym, can you spell it out first.

The full title of VALHUDES has been used to define the acronym in the text:

‘Validation of Human Papillomavirus Assays and Collection Devices for Self-samples and Urine Samples (VALHUDES)’

  • Discussion – I think there is room in the discussion to briefly talk to the importance of self-collection in reducing within-country disparities in cervical screening and cervical cancer rates, and that the design and implementation of self-sampling programs/approaches need to be under-pinned by evidence on how well methods, tests, devices, approaches, etc work in the never and under-screened populations. Using evidence from the general screen-eligible population may not lead to an equitable program, and may further widen current disparities.

We thank the reviewer for this suggestion. We certainly acknowledge that self-collection and how it can be applied in different settings, lower and middle income countries vs never- and under-screened populations in countries with mature cervical screening programs is a major factor but we do feel like it falls outside of the scope of this review which is more focused on the technical aspects of self-collection. The following sentence has been added to the Discussion to communicate this view.

‘There is a need for research into the acceptance and feasibility of using self-collection in under- or never-screened populations, but this falls outside the scope of the current review which is more focused on the technical aspects of self-collection.’

  • Throughout (eg. Line 352, Line 361) - there needs to be a period after al in et al. (Arbyn et al. was…)

This change has been executed throughout the manuscript.

  • Throughout – you sometimes hyphenate the words self-collect, self-collected, and self-collection, whereas sometimes you leave out the hyphen. APA style advises that “self-“ words should always be hyphenated.

The manuscript has been modified so that all appearances of both ‘self-collect/ed/ion’ and ‘clinician-collect/ed/ion’ are hyphenated

  • Throughout – compound adjectives preceding the term that they modify, should be hyphenated. You have not always done this, e.g. evidence-based devise (line 219). Compound adjectives that do not precede the modifying term do not need to be hyphenated. My preferences is that they are for consistency. But as long as you are consistent throughout, it does not matter. Perhaps check, line 270 follow-up, 373 flip-side, and throughout.

We thank the reviewer for this comment. We have reviewed the manuscript for compound adjectives.

  • The formatting for the in-text citations is not correct on lines 72, 272.

This formatting error has been corrected

  • Check formatting of references. E.g. reference 10 – HPV should be capitalised.

We have reviewed the references and made changes including multiple occasions of capitalisation of HPV.

Reviewer 3 Report

This is a good practical review of the evidence when is comes to choice of technology in self-collection of samples in cemical screening. I have no point to criticizes.

Author Response

This is a good practical review of the evidence when is comes to choice of technology in self-collection of samples in cervical screening. I have no point to criticize.

We thank the reviewer for their comment.

Round 2

Reviewer 1 Report

The manuscript is improved by the edits/modifications and the authors have done a good job in removing some of the unnecessary repetition